# Trypanocidal Activity of Four Sesquiterpene Lactones Isolated from Asteraceae Species

**DOI:** 10.3390/molecules25092014

**Published:** 2020-04-25

**Authors:** Orlando G. Elso, Augusto E. Bivona, Andrés Sanchez Alberti, Natacha Cerny, Lucas Fabian, Celina Morales, César A. N. Catalán, Emilio L. Malchiodi, Silvia I. Cazorla, Valeria P. Sülsen

**Affiliations:** 1Instituto de Química y Metabolismo del Fármaco (IQUIMEFA), CONICET–Universidad de Buenos Aires, Junín 956 2° floor, Buenos Aires 1113, Argentina; orlandoelso@hotmail.com (O.G.E.); lucas.fabian@gmail.com (L.F.); 2Cátedra de Farmacognosia, Facultad de Farmacia y Bioquímica, Universidad de Buenos Aires, Junín 956 2° floor, Buenos Aires 1113, Argentina; 3Cátedra de Inmunología, Facultad de Farmacia y Bioquímica, Universidad de Buenos Aires, Junín 956 4° floor, Buenos Aires 1113, Argentina; augustobivona@gmail.com (A.E.B.); andres.sanchez.alberti@gmail.com (A.S.A.); emalchio@ffyb.uba.ar (E.L.M.); 4Instituto de Estudios de la Inmunidad Humoral (IDEHU), UBA-CONICET. Junín 956 4° floor, Buenos Aires 1113, Argentina; 5Instituto de Microbiología y Parasitología Médica—CONICET (IMPaM), Facultad de Medicina, CONICET—Universidad de Buenos Aires, Paraguay 2155. 13° floor, Buenos Aires C1121ABG, Argentina; 6Instituto de Ecología y Desarrollo Sustentable (INEDES), CONICET—Universidad Nacional de Luján, Ruta 5 y Avenida Constitución, Luján 6700, Argentina; natachacerny@gmail.com; 7Departamento de Patología, Instituto de Fisiopatología Cardiovascular, Universidad de Buenos Aires, Facultad de Medicina, Buenos Aires 1113, Argentina; celinamorales2004@yahoo.com.ar; 8Instituto de Química del Noroeste—CONICET (INQUINOA), CONICET—Universidad Nacional de Tucumán, Ayacucho 471, San Miguel de Tucumán T4000INI, Argentina; ccatalan@fbqf.unt.edu.ar; 9CONICET—Centro de Referencia para Lactobacilos (CERELA), Batalla de Chacabuco 145. San Miguel de Tucumán T4000INI, Argentina

**Keywords:** *Trypanosoma cruzi*, sesquiterpene lactones, Asteraceae, *Stevia* spp., *Mikania* spp.

## Abstract

The sesquiterpene lactones eupatoriopicrin, estafietin, eupahakonenin B and minimolide have been isolated from Argentinean Astearaceae species and have been found to be active against *Trypanosoma cruzi* epimastigotes. The aim of this work was to evaluate the activity of these compounds by analyzing their effect against the stages of the parasites that are infective for the human. Even more interesting, we aimed to determine the effect of the most active and selective compound on an *in vivo* model of *T. cruzi* infection. Eupatoriopicrin was the most active against amastigotes and tripomastigotes (IC_50_ = 2.3 µg/mL, and 7.2 µg/mL, respectively) and displayed a high selectivity index. This compound was selected to study on an *in vivo* model of *T. cruzi* infection. The administration of 1 mg/kg/day of eupatoriopicrin for five consecutive days to infected mice produced a significant reduction in the parasitaemia levels in comparison with non-treated animals (area under parasitaemia curves 4.48 vs. 30.47, respectively). Skeletal muscular tissues from eupatopicrin-treated mice displayed only focal and interstitial lymphocyte inflammatory infiltrates and small areas of necrotic; by contrast, skeletal tissues from *T. cruzi* infected mice treated with the vehicle showed severe lymphocyte inflammatory infiltrates with necrosis of the adjacent myocytes. The results indicate that eupatoriopicrin could be considered a promising candidate for the development of new therapeutic agents for Chagas disease.

## 1. Introduction

Chagas disease is an infection caused by the protozoan *Trypanosoma cruzi*. The disease affects among eight million people around the world with about 100 million people being at risk. Annually, around 56,000 new vectorial and non-vectorial transmission cases are reported with 12,000 deaths [1]. Vector-borne transmission occurs in Latin America by triatomine bugs living in rural and suburban areas. The parasite can also be transmitted through organ transplant, blood transfusion and by oral and congenital routes [2]. Even though Chagas disease has been circumscribed initially to Latin America, the infection has spread worldwide due to human migration from endemic to non-endemic areas [3]. Benznidazol and nifurtimox are the only two drugs available for the treatment of Chagas disease. They are effective in the acute stage of the infection and in cases of congenital transmission. However, their efficacy diminishes as the disease progresses, being ineffective in the chronic stage. These drugs are also associated with a high non-compliance rate due to the existence of dose-dependent toxicity that leads to a discontinuation of the treatment [4]. 

The plant secondary metabolism constitutes an important source of drugs. These compounds can be used in either the original chemical form and/or after structural modifications. Among these natural compounds, terpenoids are promising leads to develop drugs to treat neglected protozoan diseases, such as Chagas disease, African trypanosomiasis, leishmaniasis and malaria [5]. Sesquiterpene lactones (STLs) are a large group of terpenic compounds mainly distributed within the Asteraceae family. These compounds present a wide chemical structural diversity and display many biological and pharmacological activities, such as antitumor, anti-inflammatory, antibacterial, antifungal, antiviral, and antiprotozoal, among others [6]. A prominent example of a STL approved for use in humans for a protozoan disease treatment is artemisinin and its derivatives, which are used for the treatment of *Plasmodium falciparum* malaria [7]. 

*Stevia* and *Mikania* genera belong to Asteraceae family. These genera typically produce sesquiterpene lactones of the guaianolide and germacranolide type, respectively. Stevia species have played an important role in therapy against infectious diseases, as they are used for skin problems, malaria, fever and colds [8]. Three main sesquiterpene lactones were isolated from Stevia maimarensis, *S. alpina*, and *S. gilleisi*, growing in northern Argentina: eupatoriopicrin, estafietin, and eupahakonenin B, respectively [9,10,11]. These compounds have shown anti-inflammatory, antitumor and antiparasitic activities, among others [12,13,14,15,16].

*Mikania* species have shown effects on the respiratory tract and central nervous system, and pharmacological activities such as anti-inflammatory and antimicrobial, among others. In this sense, antibacterial, antiparasitic and antifungal activities have been described for some *Mikania* species, which have been associated with the presence of sesquiterpene lactones and diterpenoids [17]. *M. minima* is an Argentinean species from which several sesquiterpene lactones have been isolated, being minimolide the main germacranolide present in this plant [18].

In a previous work, we reported the activity of eupatoriopicrin, estafietin, eupahakonenin B and minimolide, isolated from *Stevia maimarensis*, *S. alpina*, *S. gilleisi* and *Mikania minima*, respectively, on *T. cruzi* epimastigotes [16]. The aim of this work was to further characterize the effect of these compounds by evaluating their activity against trypomastigotes and amastigotes and most importantly to carry out a proof of concept study by treating *T. cruzi* infected animals with the most active compound. 

## 2. Results

### 2.1. Isolation and Identification of the Sesquiterpene Lactones from Asteraceae Species

The sesquiterpene lactones eupatoriopicrin, estafietin, eupahakonenin B and minimolide were isolated from the organic extracts of *Stevia maimarensis*, *S. alpina*, *S. gilliesii* and *Mikania minima*, with yields of 0.18%, 0.63%, 0.27% and 0.50%, respectively (Figure 1). The four compounds were identified by spectroscopic methods and by comparison with literature data [9,10,11,18]. It is worth noting here that in ref. [11] the 1 (10) double bond of estafietin is misplaced; it should be placed in position 10 (14) as shown in Figure 1. 

### 2.2. In Vitro Anti-T. cruzi Activity

The trypanocidal effect of the STLs was assessed on bloodstream trypomastigotes by counting the remaining live parasites in a Newbauer chamber after incubation with each compound. The four STLs displayed trypanocidal activity on the bloodstream form of *T. cruzi* with 50% inhibitory concentration (IC_50_) values of 7.2 ± 0.3 µg/mL for eupatoriopicrin, 28.9±4.1 µg/mL for estafietin, 11.9 ± 4.5 µg/mL for eupahakonenin B and 7.7 ± 0.4 µg/mL for minimolide (Figure 2). The IC_50_ of the reference drug benznidazole was 16.4 ± 2.3 µg/mL. 

The effect of the compounds on the intracellular form of *T. cruzi* was determined by a spectrophotometric method. Eupatoriopicrin was the most active STL on *T. cruzi* amastigotes with an IC_50_ value of 2.3 ± 0.5 µg/mL. The IC_50_ values for minimolide, eupahakonenin B and estafietin were 9.2 ± 0.8, 32.2 ± 4.6 and 26.9 ± 4.3 µg/mL, respectively (Figure 3). Benznidazole presented an IC_50_ of 3.5± 0.6 µg/mL.

### 2.3. Cytotoxicity

The cytotoxicity of the four STLs was evaluated on Vero cells by the MTT assay (Figure 4). The 50% cytotoxicity concentration (CC_50_) value for eupatoriopicrin was 93.3 ± 1.2 µg/mL, with selectivity indexes (SI) of 12.9 and 40.6 for *T. cruzi* trypomastigotes and amastigotes, respectively (Table 1). Estafietin, eupahakonenin B and minimolide presented CC_50_ values of 197.0 ± 24.3, 123.9 ± 6.2 and 98.9 ± 2.8 µg/mL, respectively. Selectivity indexes higher than five were obtained for the three STLs on trypomastigotes and amastigotes (Table 1). Benznidazole showed a CC_50 of_ of 304.8 ± 16.1 µg/mL. 

### 2.4. In Vivo Assay

Since eupatoriopicrin was the most active and selective STL against *T. cruzi* trypomastigotes and amastigotes, this compound was selected for evaluation on an *in vivo* model of Chagas disease. Balb/c mice were infected with *T. cruzi* trypomastigotes (K98 strain) and treated with benznidazole, eupatoriopicrin or the vehicle (DMSO) as control. Each compound was administered for five consecutive days since day 11 post infection (day in which parasites became detectable by microhemotocrit analysis). Mice receiving the STL presented lower parasitaemia levels, as compared to untreated animals (Figure 5A). Parasitaemia was monitored till day 48 post-infection when parasites in mice blood became undetectable. *T. cruzi* infected mice treated with eupatoriopicrin showed an important decrease in the area under the parasitaemia curve (AUC) regarding the control group (AUC: 4.48 ± 1.36 and 30.47 ± 7.15, respectively; *p* < 0.001) (Figure 5B). Similar results were observed for the reference drug, Benznidazole which showed a 34.8 times reduction in the AUC with respect to the control (AUC: 0.87 ± 0.77).

### 2.5. Histopathological Analysis

The anti-*T. cruzi* activity of the STL eupatoriopicrin was also analyzed on the chronic phase of the parasite infection. Samples of heart and skeletal (quadriceps) muscles from the experimental mice were taken and stained with hematoxylin-eosin for the histological examination. No significant differences were observed in the architecture of the hearts form the different experimental animals. Small foci of interstitial or perivascular lymphocyte infiltrate without damage to adjacent myocytes were observed in both control and treated mice (Figure 6A–C). By contrast, significant changes were observed when skeletal muscle from the experimental mice were analyzed. The skeletal muscle from *T. cruzi* infected mice treated with the vehicle (DMSO) showed interstitial coalescent lymphocyte inflammatory infiltrates with necrosis of the adjacent myocytes. By contrast, muscle tissues from eupatopicrin-treated mice displayed only focal and interstitial lymphocyte inflammatory infiltrates and small areas of necrotic tissue (Figure 6D–F).

## 3. Discussion

In a previous study, we reported the anti-*T.cruzi* activity of the STLs eupatoriopicrin, estafietin, eupahakonenin B and minimolide against *T. cruzi* epimastigotes [8]. The four sesquiterpene lactones have been isolated from *S. maimarensis*, *S. alpina*, *S. gilliesi* and *M. minima*, respectively. The compounds were identified by spectroscopic methods. The four compounds have previously shown significant activity on *T. cruzi* epimastigotes. Although the epimastigote stage of the parasite is non-infective, it is easy to cultivate and, therefore, useful for a preliminary screening test. Here, we decided to evaluate the effect of the STLs on the bloodstream and intracellular forms of *T. cruzi*. All the tested compounds exerted trypanocidal activity against trypomastigotes, with IC_50_ values ranging from 7.2 to 28.9 µg/mL. Besides, when the compounds were tested on amastigotes, eupatoriopicrin showed the highest inhibitory effect (IC_50_ = 2.3 ± 0.5 µg/mL). 

In order to determine the selectivity of these compounds an *in vitro* cytotoxicity assay was carried out on Vero cells. The STLs rendered SI values ranging from 6.8 to 12.9 on bloodstream tripomastigotes. Eupatoriopicrin also displayed significant selectivity for intracellular amastigotes in comparison with the other tested compounds (SI = 40.6). Eupatoriopicrin has also been found to be active against *Trypanosoma brucei rhodesiense*, with an IC_50_ of 0.43 µg/mL but with low selectivity [15]. The trypanocidal activity of estafietin on the infective and intracellular forms of *T. cruzi* has been previously described by our group [19]. This compound was included in the in vitro assays carried out in this work in order to compare its activity with that of the other three STLs. Eupatoriopicrin was the most active and selective compound, therefore it was selected for in vivo assessment in *T. cruzi* infected mice.

The clinical outcome of Chagas disease is highly variable, mainly because of the heterogeneity of its etiologic agent [20]. *Trypanosoma cruzi* is a monophyletic but heterogeneous group conformed by several Discrete Typing Units (DTUs) named TcI to TcVI [21]. The *in vitro* trypanocidal efficacy of the STLs was evaluated against two highly virulent *T. cruzi* strains, RA and Tulahuen, which belong to TcII-DTU and TcVI-DTU, respectively. By contrast, for the in vivo study we selected the K98 clone (TcI-DTU) because it does not present lethality during the acute phase [22]. This model mimics better the human infection, in which a small percentage (1–2%) of patients die from acute Chagas [23], and most of the patients progress to a chronic infection [24]. 

The administration of 1 mg/kg/day of eupatoriopicrin for five consecutive days to *T. cruzi*-infected mice induced a significant reduction in the parasitaemia levels in comparison with non-treated control animals. More interestingly, this reduction was similar to that achieved with the reference drug, benznidazole. Eupatoriopicrin showed a potent effect, causing a greater decrease in the number of circulating parasites. Although reductions in the parasitaemia levels have been reported for others sesquiterpene lactones, scarce studies have investigated their performance to prevent tissue damages during the chronic phase of the parasite infection [25,26,27,28]. Importantly, we found that the treatment of infected animals with eupatoriopicrin was also effective to reduce the cardiac and mainly the skeletal damage associated to *T. cruzi* infection. 

## 4. Materials and Methods

### 4.1. Plant Material

The aerial parts of *Stevia maimarensis* (Hieron.) Cabrera (Asteraceae) were collected in the province of Jujuy, Argentina, in March 2017. The aerial parts of *Stevia alpina* Griseb. (Asteraceae) and *Stevia gilliesii* Hook. & Arn. (Asteraceae) were collected in the province of Catamarca, Argentina, in March 2015. The aerial parts of *Mikania minima* (Baker) B.L. Rob. (Asteraceae) were collected in the province of Tucumán, Argentina, in March 2015. The plant material was identified by Hernan Bach PhD and voucher specimens were deposited at the Museo de Farmacobotánica, Facultad de Farmacia y Bioquímica, Universidad de Buenos Aires under the numbers BAF 12264, BAF 12266, BAF 12267 and BAF 12268, respectively.

### 4.2. Sesquiterpene Lactones Isolation

Fifty grams of dried aerial parts of each plant species were extracted twice at room temperature with dichloromethane (DCM) (500 mL, 3 min.). Filtrates were concentrated on a rotatory evaporator at 40 °C under reduced pressure. *S. maimarensis* and *S. gilliesii* crude extracts were dissolved in warm ethanol (140 mL), the solution was placed in a dropping funnel, water (60 mL) was added (ratio ethanol–water 70:30 *v/v*) and the mixture was extracted three times with hexane (3 × 50 mL) and then with DCM (3 × 50 mL). DCM sub-extracts were reunited and evaporated to yield dewaxed extract, rich in STLs.

*S. alpina* and *M. minima* crude extracts and *S. maimarensis* and *S. gilliesii* dewaxed extracts were fractionated by silica gel column chromatography (50 × 4.5 cm, 180 g, 230–400 mesh) and eluted isocratically with dichloromethane-ethyl acetate mixtures: 1:2 for *S. maimarensis* and *S. gilliesii*, 9.5:0.5 for *S. alpina* and 2:1 for *M. minima*. Column chromatography eluates were monitored by thin layer chromatography (TLC) using STL standards. Fractions showing a single spot corresponding to the desired compounds were pooled and dried in a rotary evaporator. Fractions enriched in the desired compound were also pooled and rechromatographed as above to recover additional amounts of lactone.

The STL eupatoriopicrin precipitated from *S. maimarensis* fractions as white crystals. The crystals were washed twice with a 7:3 mixture of ethyl ether-ethyl acetate. Recrystallization from *n*-heptane-ethyl acetate yielded needles mp 158–160 °C [9]. Eupahakonenin B was obtained as a greenish gum when *S. gilliesi* fractions showing a single spot on TLC with R_f_ identical to a reference standard [10] were reunited and evaporated in rotavapor. Less pure fractions were reunited and rechromatographed under the same conditions described above in order to increase the amount of the isolated compound. The rechromatographed lactone was isolated as a colorless gum. Estafietin and minimolide were purified by column chromatography from crude extracts of *S. alpina* and *Mikania minima*, respectively (See Appendix A). In both cases, STLs-rich fractions were pooled, the solvent was evaporated in a rotary evaporator and the residue was suspended in a minimum volume of warm heptane-ethyl acetate (1:2) to near complete dissolution and then kept in the refrigerator overnight. Both STLs precipitated as white needles and were recrystallized from heptane-ethyl acetate; estafietin, mp 104–105 °C [11]; minimolide, mp 112–113 °C [18].

The purity of the compounds was determined by HPLC using a RP-C18 column (250 × 4.6 mm - 5 µm mesh) and a water: acetonitrile gradient (35% to 95% acetonitrile in 30 min. for eupatoriopicrin, minimolide and eupahakonenin B and 45% to 95% acetonitrile in 30 min. for estafietin. A flow rate of 1 mL/min was used. The diode array detector was set at 210 nm. The purity of the sesquiterpene lactones determined by HPLC was: eupatoriopicrin: 94.6%, minimolide: 99.8%, estafietin: 93.3% and eupahakonenin B: 93.9%. Further recrystallization of eupatoriopicrin, estafietin and eupahakonenin B yielded analytical samples with a purity >98.2% in all three cases. The identity of the compounds was confirmed by UV, IR, NMR and MS spectroscopy. 

### 4.3. Parasites

*Trypanosoma cruzi* bloodstream trypomastigotes from RA and K98 strains [22] as well as transfected trypomastigotes expressing β galactosidase [26], were obtained from infected CF1 mice by cardiac puncture at the peak of parasitaemia on day 15 post infection. 

Transfected trypomastigotes (Clone C4 of the Tulahuen strain) that express the *Escherichia coli* beta-galactosidase gene were kindly provided by Dr. Buckner [29].

### 4.4. Activity Assay on Bloodstream T. Cruzi Trypomastigotes

Mouse blood containing trypomastigotes from the RA *T. cruzi* strain was diluted with an RPMI medium to a cell density of 1.5 × 10^6^ parasites /mL. Parasites were then seeded in duplicate in a 96-well microplate with increasing concentrations of each STL (0–100 µg/mL). Additional controls in which parasites were incubated with the solvent were also included. After 24 h incubation at 4 °C, parasites were diluted 1/10 with lysis buffer and the remaining live trypomastigotes were counted in a Neubauer chamber. The percentage of live trypomastigotes was calculated as {[(live parasites after incubation)/(live parasites in untreated wells)] × 100}. 

### 4.5. Activity Assay on Intracellular T. Cruzi Amastigotes

A 96-well plate was seeded with the non-phagocytic Vero cell line at 5 × 10^3^ per well in 100 μL of culture medium and incubated for 2 h at 37 °C in a 5% CO_2_ atmosphere. Cells were washed and infected with transfected Tulhauen bloodstream trypomastigotes expressing β-galactosidase (MOI:1/10 cells/parasites). After 24 h of co-culture, the plate was washed twice with PBS to remove extracellular parasites and each compound and benznidazole was added at 0–100 μg/mL per well in 150 μL of fresh complete RPMI medium without phenol red. Non-treated cells (100% infection) and non-infected cells (0% infection) were used as controls. After five days, cells were lysed with 1% Nonidet P-40 and chlorophenol red-β-d-galactopyranoside (CPRG) (100 μM) was added as β-galactosidase substrate. After 4–6 h incubation at 37 °C, the absorbance was measured at 570 nm in a microplate reader. The percentage of inhibition was calculated as 100 − {[(absorbance of treated infected cells)/(absorbance of untreated infected cells) × 100}.

### 4.6. Cytotoxicity

Cytotoxicity was determined by the MTT method. Briefly, Vero cells (5 × 10^5^ c/mL) were settled at a final volume of 150 µl in a flat-bottom, 96-well microplate and cultured at 37 °C in a 5% CO_2_ atmosphere in the presence of increasing concentrations of the pure compounds and the reference drug benznidazole. Cells incubated in absence of the STLs were used as a 100% viability control. After 24 h, 3-(4,5-dimethylthiazol-2-yl)-2,5-diphenyltetrazolium bromide (MTT) was added at a final concentration of 1.5 mg/mL. Plates were incubated for two additional hours at 37 °C. The purple formazan crystals were completely dissolved by adding 150 µl of ethanol and the absorbance was read at 595 nm in a microplate reader. Results were expressed as the ratio between the absorbance in the presence and absence of the compound multiplied by 100.

### 4.7. In Vivo Trypanocidal Activity

Inbred Balb/c mice (male, eight weeks old, and 26 g average weight) were nursed at Departamento de Microbiología, Facultad de Medicina, Universidad de Buenos Aires. Mice were infected with 3 × 10^5^
*T. cruzi* trypomastigotes of the K98 strain by intraperitoneal route. Parasitaemia was measured weekly. Blood samples were diluted 1:5 in lysis buffer (0.75% NH_4_Cl, 0.2% Tris, pH 7.2) and parasites were counted in a Neubauer chamber. 

Mice were divided in groups of five animals each and the drugs were administered by the intraperitoneal route (1 mg/kg of body weight/day) for five consecutive days after infection (11 to 15 dpi). Eupatoriopicrin and benznidazol were diluted with DMSO and the concentration was adjusted with 0.1M phosphate buffered saline (pH 7.2). A vehicle was employed as a negative control. 

### 4.8. Histopathological Analysis

Samples of heart and skeletal (quadriceps) muscles were dissected at 100 dpi and fixed with 4% formalin in PBS. The fixed tissue was embedded in paraffin, sectioned and stained with haematoxylin and eosin. Sections were analyzed at 200 and 400× magnifications. Inflammatory infiltrates were qualitative graded as focal, multiple non-confluent, multiple confluent and multiple diffuse infiltrate [30]. The presence of necrotic areas was evaluated as well.

### 4.9. Ethic Statement 

Animal experiments were approved by the Review Board of Ethics of Universidad de Buenos Aires, Facultad de Medicina, Argentina (no. 2943/2013) and conducted in accordance with the Guide for the Care and Use of Laboratory Animals of the National Research Council.

### 4.10. Statistical Analysis

Results are presented as means ± SEM. The GraphPad Prism 6.0 software (GraphPad Software Inc., San Diego, CA) was employed to carry out calculations. Parasitaemia levels was analyzed by the Dunett’s test. The statistical significance was determined by one-way analysis of variance (ANOVA) performed with the GraphPad Prism 6.0 software. Comparisons were referred to the control group. *p* values <0.05 were considered significant. 

## 5. Conclusions

Eupatoriopicrin, estafietin, eupahakonenin B and minimolide showed an in vitro anti-*T. cruzi* activity against the infective and intracellular forms of *T. cruzi*. Besides, the STL eupatoriopicrin, isolated from *Stevia maimarensis*, was also effective in an in vivo model of Chagas disease, reducing the parasiteamia and the tissue damage of the skeletal muscle. Even more important, the high anti-parasite activity of this STL against strains of *T. cruzi* with different genetic backgrounds signals eupatoriopicrin as a promising candidate for further studies in the search of novel therapeutic agents for Chagas disease.

## Figures and Tables

**Figure 1 molecules-25-02014-f001:**
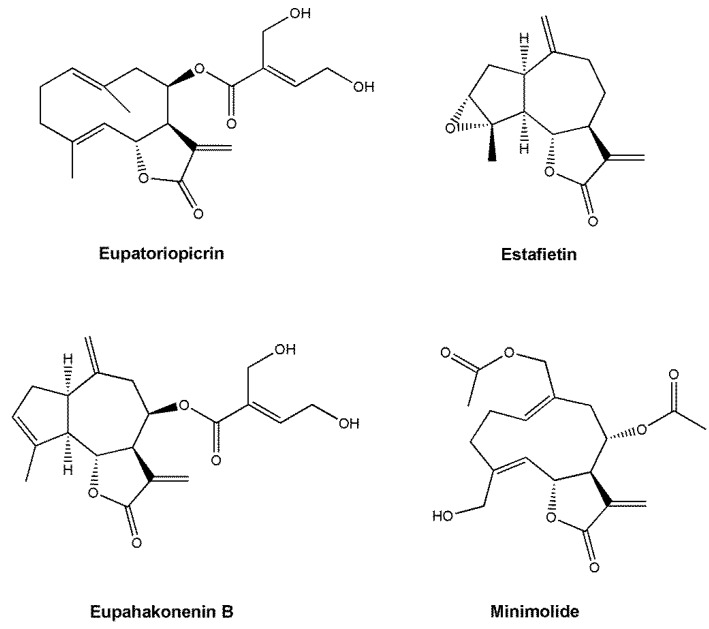
Chemical structures of eupatoriopicrin, estafietin, eupahakonenin B and minimolide.

**Figure 2 molecules-25-02014-f002:**
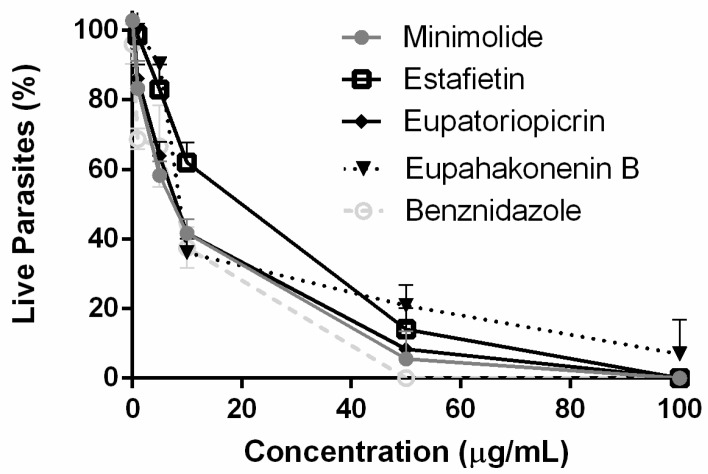
Trypanocidal activities of eupatoriopicrin, estafietin, eupahakonenin B and minimolide against *T. cruzi* trypomastigotes. Bloodstream trypomastigotes (RA) were cultured in duplicate in the presence of 0 to 100 µg/mL of eupatoriopicrin, estafietin, eupahakonenin B and minimolide. Cultures were done in 96-well plates with 1.5 × 10^6^ parasites/mL during 24 h, and the remaining live parasites were counted in a Neubauer chamber. Results are expressed as the mean ± SEM of three independent experiments.

**Figure 3 molecules-25-02014-f003:**
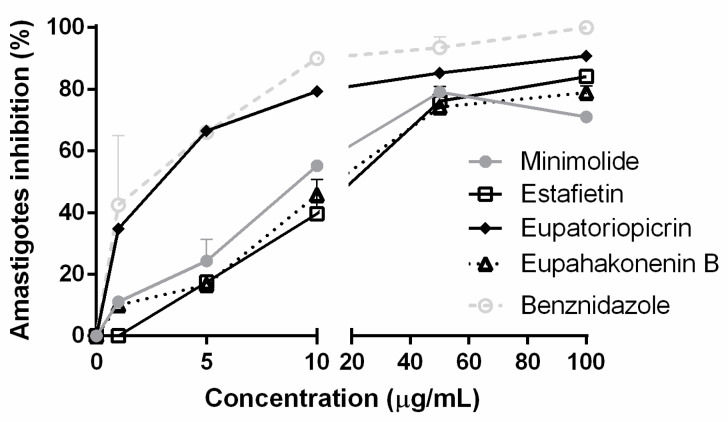
Inhibition of *T. cruzi* amastigotes by eupatoriopicrin, estafietin, eupahakonenin B and minimolide. Vero cells (5 × 10^3^ cells/well) were seeded in 96-well plate and infected 24 h later with transfected trypomastigotes expressing β-galactosidase. After 24 h of co-culture, plates were washed, and compounds were added at 0–100 µg/mL in 150 µl medium. On day six post-infection, the reaction was developed by the addition of CPRG (100 mM) and Nonidet P-40 (1%). Plates were incubated for 6 h and the absorbance was read at 570 nm. Infected, non-treated cells were used as 100% infection control. The percentage of inhibition was calculated as 100 − {[(Absorbance of treated infected cells)/(Absorbance of untreated infected cells)] × 100}.

**Figure 4 molecules-25-02014-f004:**
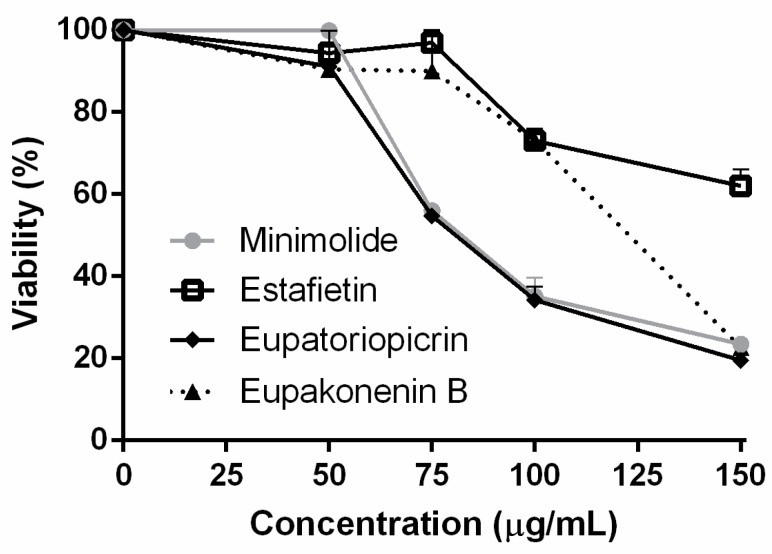
In vitro cytotoxicity of eupatoriopicrin, estafietin, eupahakonenin B and minimolide on Vero cells. Cells were incubated for 24 h with increasing concentrations of the compounds. Cell viability was determined by the MTT method. Results are expressed as percentage of viability. Bars represent the means ± SEM of three experiments carried out in duplicate.

**Figure 5 molecules-25-02014-f005:**
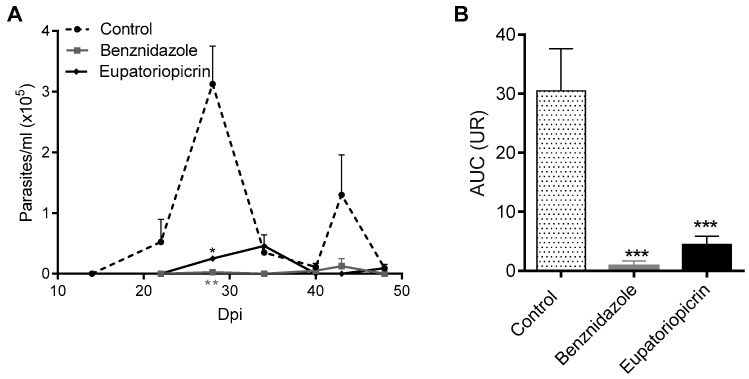
In vivo anti-*T. cruzi* activity of eupatoriopicrin. Balb/c mice infected with 3 × 10^5^
*T. cruzi* trypomastigotes (K98 strain) were treated for five consecutive days (days 11 to 15 post-infection) with eupatoriopicrin, beznidazole or DMSO (vehicle, as control). Parasitaemia (**A**) and the area under parasitaemia curve (AUC) (**B**), were determined in experimental mice. Results expressed as mean ± SEM, are representative of three independent experiments. One-way ANOVA and Dunnett’s post-test *** *p* < 0.001.

**Figure 6 molecules-25-02014-f006:**
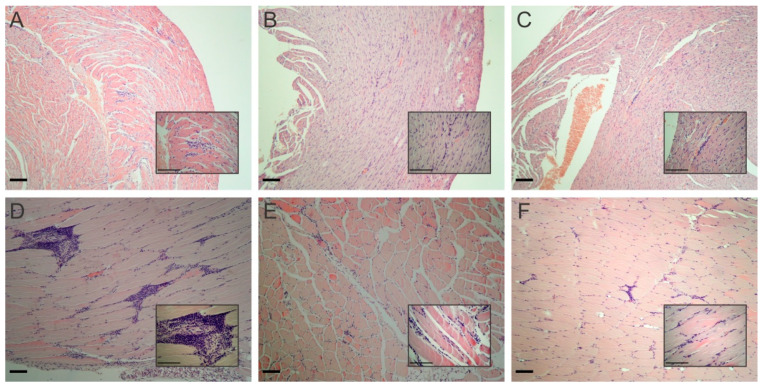
Histopathological analysis of heart tissue (upper panels) and muscular tissue, quadriceps, (lower panels), from experimental mice. Balb/c mice infected with *T. cruzi* and treated with vehicle (**A** and **D**); benznidazole (**B** and **E**) and eupatoriopicrin (**C** and **F**), were sacrificed on 100 dpi and a histopathological analysis of the tissues was performed using haematoxylin and eosin staining. Magnification 200× and 400×. Scale bar is 100 μm.

**Table 1 molecules-25-02014-t001:** Selectivity index of eupatoriopicrin, estafietin, eupahakonenin B and minimolide for trypomastigotes and amastigotes.

Sesquiterpene Lactones	Selectivity Index (SI)	
	Trypomastigotes	Amastigotes
Eupatoriopicrin	12.9	40.6
Estafietin	6.8	7.3
Eupahakonenin B	10.4	3.8
Minimolide	12.8	10.7
Benznidazole	18.6	87.1

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
