# Peer review of "Trypanocidal Activity of Four Sesquiterpene Lactones Isolated from Asteraceae Species"

_molecules, 2020, doi:10.3390/molecules25092014_

Round 1

Reviewer 1 Report

the manuscript: Trypanocidal activity of four sesquiterpene lactones isolated from Asteraceae species is quite interesting and well-written. However, certain points required more attention.

Among these points:

1- In the introduction part more details are required regarding the plants of interest from which the sesquiterpenes are isolated to shed light about possible anti-infective activity 

2- The identification method and voucher numbers are missed

3- A schematic diagram for the isolation as a supplementary will be of great value

4- Positive control is missed in many in vitro testing experiments

Author Response

Reviewer 1

In the introduction part more details are required regarding the plants of interest from which the sesquiterpenes are isolated to shed light about possible anti-infective activity

The introduction has been completed according to the suggestion of the reviewer.

The identification method and voucher numbers are missed

The information required is included in the Material and Methods section, point 4.1. Plant Material. The paragraph has been corrected for better understanding.

A schematic diagram for the isolation as a supplementary will be of great value

A schematic diagram detailing the isolation of each compound has been included as supplementary material.

Positive control is missed in many in vitro testing experiments.

The required information has been added in Figures 2 and 3.

The in vitro cytotoxic effect of benznidazole on Vero cells was evaluated by the MTT method and was expressed as percentage of cell viability (CC50=304.8±16.1 µg/mL). Based on this result and the IC50 values, the selectivity indexes were added in Table 1.

Reviewer 2 Report

In my opinion, the work presented by the authors is of very good quality and deserves to be published in molecules.
I would only like to make a few minor scopes to the authors. In the first place, it would be interesting if they not only show the graphs associated with the activities they indicate, but also integrate a table where the error associated with each of these measurements is shown, this would give an added value to their results, since if the associated error is small we could be in the presence of possible alternatives for the treatment of Chagas disease or other similar ones.
What is also not entirely clear to me is whether the authors tested the toxicity of eupatoriopicrin in the in vivo experiments. I would have expected a control group to only be treated with this compound, without being contaminated with trypomastigotes. This is essential to know if the compound can be evaluated as a possible drug in the future.
Finally and I declare my ignorance about it. Selectivity indices do not have an associated error percentage?
Once the authors answer the questions raised, the work may be published in molecules.

Author Response

Reviewer 2

In my opinion, the work presented by the authors is of very good quality and deserves to be published in molecules. I would only like to make a few minor scopes to the authors. In the first place, it would be interesting if they not only show the graphs associated with the activities they indicate, but also integrate a table where the error associated with each of these measurements is shown, this would give an added value to their results, since if the associated error is small we could be in the presence of possible alternatives for the treatment of Chagas disease or other similar ones.

In the Figures Mean ± SEM of each point was plotted with respect to the concentration of the compounds. We consider that it is not necessary to include a table which in terms would give the same information. But to be more accurate the IC50 and CC50 values were expressed as the Mean±SEM in contrast to the previous manuscript in which results were expressed as the Mean.

What is also not entirely clear to me is whether the authors tested the toxicity of eupatoriopicrin in the in vivo experiments. I would have expected a control group to only be treated with this compound, without being contaminated with trypomastigotes. This is essential to know if the compound can be evaluated as a possible drug in the future.

Toxicity of eupatoriopicrin was only tested in vitro. But its evaluation on an in vivo assay is a very interesting point that we will carried out in the near future together with the evaluation of the possible mechanisms by which the compound display its anti-T. cruzi activity.

Finally and I declare my ignorance about it. Selectivity indices do not have an associated error percentage?

Selectivity indexes are calculated as the relation between the CC50 and the IC50, not presenting an error associated.

Reviewer 3 Report

The presented study is another one from the series of papers, written by an experienced Argentinean team, that deals with antiprotozoan activity of sesquiterpene lactones from the Asteraceae. The results are interesting and the paper is well written. However, some corrections are necessary before the publication. Some additional information should also be included in the final version of the manuscript.

Figure 1 – the structure of estafietin should be corrected (incorrect position of the double bond: 10(1) instead of 10(14)).

Figures 2-4 and Table 1 – the strain of trypomastigotes used should be specified in the legends (RA or Tulahuen, see Discussion section, line 24).

Table 1 – The selectivity index of Beznidazole should be also given.

Figure 5 – what was the dose of trypomastigotes used to induce infection?

Page 8, section 4.2., line 20 – what about the purity (%) of the isolated compounds?

Page 9, section 4.3. - please, include an information on the provenience of the amastigotes and Tulahuen strain of trypomastigotes used in the experiments.

Page 9, section 4.3., lines 1 and 3 – please convert the reference style into the appropriate format.

Page 10, section 4.8., line 5 – “(Martin et al., 2007)” should be deleted.

Reference [23] – erroneous volume number and pagination.

Author Response

Reviewer 3

Figure 1- the structure of estafietin should be corrected (incorrect position of the double bond: 10(1) instead 10(14)).

The structure of estafietin has been corrected. In reference 10, De Heluani et al., Guaianolides, heliangolides and other constituents from Stevia alpina. Phytochemistry 1989, 28, 1931-1935; the 1(10) double bond of estafietin is misplaced. A sentence making reference to this mistake has been included in the manuscript in the Results section, 2.1. Isolation and identification of the sesquiterpene lactones from Asteraceae species.

Figures 2-4 and Table 1- the strain of trypomastigotes used should be specified in the legends (RA or Tulhauen, see Discussion section line 24.

This information was added as suggested.

Table 1 – The selectivity index of Benznidazol should be also given

Selectivity index of benznidazol has been included in Table 1.

What was the dose of trypomastigotes used to induce infection?

As has been mentioned in the material and methods section, mice were infected with 3x105 T. cruzi trypomastigotes of the K98 strain by intraperitoneal route. This information was also added in Figure 2 legend.

Page 8, section 4.2, line 20- what about the purity (%) of the isolated compounds?

The purity of the sesquiterpene lactones was determined by HPLC: eupatoriopicrin: 94.6%, minimolide: 99.8%, estafietin: 93.3% and eupahakonenin B: 93.9%. Further recrystallization of eupatoriopicrin, estafietin and eupahakonenin B yielded analytical samples with a purity >98.2% in all three cases. This information has been included in the manuscript.

Page 9, section 4.3 – please, include an information of the provenience of the amastigotes and Tulahuen strain of trypomastigotes used in the experiments.

The information has been added as required

Page 10, section 4.8, line 5 – “Martin et al., 2007” should be deleted

Has been done

Reference [23]-erroneous volume number and pagination

The volume and pagination of the reference have been corrected.

Round 2

Reviewer 1 Report

The manuscript has much improved but still the similarity index is an obstacle